



# Air pollution monitoring: Development of ammonia (NH₃) dynamic reference gas mixtures at nmol/mol levels for improving the lack of traceability of measurements

Tatiana Macé[1], Maitane Iturrate-Garcia[2], Céline Pascale[2], Bernhard Niederhauser[2], Sophie Vaslin-Reimann[1], Christophe Sutour[1]

[1]Department of Gas Metrology, Laboratoire National de Métrologie et d'Essais LNE, Paris Cedex 15, 75724, France
[2]Department of Chemical and Biological Metrology, Federal Institute of Metrology METAS, Bern-Wabern, CH-3003, Switzerland

*Correspondence to*: Tatiana Macé (Tatiana.mace@lne.fr)

**Abstract.** The measurement of ammonia (NH₃) in ambient air is a sensitive and priority topic due to its harmful effects on human health and ecosystems. NH₃ emissions have continuously increased over the last century in Europe, because of intensive livestock practices and enhanced use of nitrogen-based fertilizers. European air quality monitoring networks monitor atmospheric NH₃ amount fractions. However, the lack of stable reference gas mixtures (RGMs) at atmospheric amount fractions to calibrate NH₃ analyzers is a common issue of the networks, which results in data that are not accurate, traceable and, thus, geographically comparable. In order to cover this lack, LNE developed, in close collaboration with the company 2M PROCESS, a gas reference generator to generate dynamically NH₃ RGMs in air. The method is based on gas permeation and further dynamic dilution to obtain an amount fraction range between 1 and 400 nmol/mol. The calibration of the elements of the generator against LNE primary standards ensures the traceability of the RGMs to the international system of units. Furthermore, the highly accurate flow and oven temperature measurements of the reference generator, together with the associated calibration procedure defined by LNE, guarantee relative expanded uncertainties of the calibration of the NH₃ analyzers lower than 2 % (coverage factor =2). This result is very satisfactory considering the low NH₃ amount fraction levels (1 to 400 nmol/mol) and the phenomena of adsorption and desorption, especially in the presence of traces of water on the surfaces in contact. A bilateral comparison was organized between METAS and LNE, consisting on the calibration of a PICARRO G2103 gas analyzer by both national metrology institutes (NMI). The results highlighted the good agreement between the NH₃ reference generators developed by the two institutes and allowed to validate both LNE's reference generator and calibration procedure. The development of the NH₃ reference generator has already raised great interest within the French air quality monitoring networks (AASQA). Since the end of 2020, LNE calibrated several NH₃ analyzers of the networks. These requests shows the interest of the AASQA in the development of this new gas reference generator to guarantee the traceability of measurements carried out on the French territory.



# 1 Introduction

Measuring ammonia ($NH_3$) in ambient air is a sensitive and priority topic because of its harmful effects on human health and ecosystems. Ammonia is the main precursor of atmospheric acid neutralization and therefore affects the long-range transport

of sulfur dioxide ($SO_2$) and nitrogen oxide(s) ($NO_x$). Moreover, atmospheric $NH_3$ contributes to the formation and stabilization of suspended particles ($PM_{10}$, $PM_{2.5}$ and aerosols) (Holmes, 2007), which have adverse human health effects such as premature deaths. In 2016, estimations suggested 4.2 million premature deaths worldwide caused by suspended particles (WMO, 2018). In addition, the deposition of ammonia on particle-borne molecules contributes to the eutrophication and acidification of soils and fresh water (Schöpp et al., 2003; Pinder et al., 2008) and thus to the degradation of soil and

water quality. These phenomena therefore have negative effects on biodiversity and ecosystems.

Main sources of atmospheric $NH_3$ are anthropogenic, such as fossil fuel combustion, power plants, vehicular exhaust and agricultural activities (Behera et al., 2013; Wu et al., 2020). The latter accounts for about 60 % of the total $NH_3$ global emissions (Clarisse et al., 2009; Paulot et al., 2014). Organic emissions from livestock, applications of organic, mineral and organo-mineral fertilizers on crops are the main agricultural sources of $NH_3$. In Europe, the European Union (EU)

agricultural sector accounts for 93% of European $NH_3$ emissions (EEA, 2020). Despite the management strategies adopted by EU, $NH_3$ emissions have continuously increased (Aksoyoglu et al. 2020; Warner et al., 2017), mainly due to the development of intensive agricultural practices and the associated use of nitrogen-based fertilizers.

Air pollution spreads across national borders and over long distances. This is why ammonia was included in the 1999 Gothenburg Protocol, revised in 2012 (UNECE, 2012), which is part of the Convention on Long-range Transboundary Air

Pollution (United Nations, 1999). The protocol was established with the aim of reducing acidification, eutrophication and ozone concentrations by setting emission ceilings to be met by 2010. Following the revision of the Gothenburg Protocol, the Directive (EU) 2016/2284 (European Commission, 2016), defines national emission ceilings for each Member State, in particular for $NH_3$. As $NH_3$ is recognized as a precursor of secondary inorganic aerosols, controlling its emissions is important to reduce $PM_{2.5}$ and $PM_{10}$ concentrations (Holmes, 2007; Clappier et al., 2021). Bessagnet et al. (2014), who used

three different chemistry transport models, showed an underestimation of the formation of ammonium particles (e.g. ammonium nitrate or sulfate) and concluded that $NH_3$ had a much greater role than initial estimates. Contrary to the emission ceilings, no limit or threshold value is defined for $NH_3$ in ambient air in the European Directives. With regard to the protection of ecosystems, particularly of the vegetation, $NH_3$ concentration Critical Levels (CLEs) ─ based on measurements and observations ─ were set by the International Cooperative Programme (ICP) Vegetation of the United Nations Economic

Commission for Europe (UNECE) in 1988. The updated $NH_3$ CLE for the protection of lichens is 1 µg/m$^3$ and 3 µg/m$^3$ for higher order plants (annual means under real conditions). A monthly critical level of 23 µg/m$^3$ was also provisionally adopted to take into account possible high peaks in emissions during periods of manure application (e.g. in spring) (Cape et al., 2009). Despite the importance of measuring ammonia for monitoring ambient air quality, neither the Directive (EU) 2016/2284 nor the Ambient Air Quality Directive 2008/50/EC (European Commission, 2008) provides recommendations for





reliable measurements. In particular, recommendations on method selection, instrument calibration (procedures, frequencies, reference gas mixtures (RGMs) to be used, etc.), maximum allowed expanded uncertainty, quality assurance and quality control procedures (QA/QC), as well as guidelines on an infrastructure to ensure the metrological traceability of measurements are lacking.

Available $NH_3$ measurement techniques are very diverse and include direct and indirect methods, with and without
sample preconcentration steps. Direct methods of making real-time measurements are based on spectroscopic techniques such as cavity ring-down spectroscopy (CRDS) (Martin et al., 2016). Indirect methods consist of using passive samplers that are exposed to ambient air for a certain period of time. During this period, $NH_3$ is collected on a support impregnated with reagent (acid-base reaction), which is then analyzed after extraction by ion chromatography or spectrometry (Tang et al., 2001). The selection of an appropriate method generally depends on the concentration levels to be measured in ambient air,
the desired spatial and temporal representativeness and the restrictions to be taken into account (e.g. installation in the field, investment and operating costs). Since $NH_3$ is not regulated in ambient air, these methods are often poorly characterized and they do not allow reliable measurements taking into account effective quality assurance and quality control procedures (QC/QA) (calibration, traceability, maximum allowed expanded uncertainty, etc.). For example, direct methods based on spectroscopic techniques are characterized by metrological performances that vary significantly among themselves,
especially in terms of accuracy and resolution. In addition, the sampling systems and the control of relative humidity − key points in ammonia measurements − are not adequate. Furthermore, the lack of traceability of the measurements to the International System of Units (SI-traceability) performed with these instruments is highlighted, as stable RGMs in gas cylinders at low amount fractions (i.e. ambient air level; 0 – 500 nmol/mol) are not commercially available. Reasons for that are the reactivity of ammonia with the inner walls of the cylinder and the phenomena of adsorption and desorption,
especially in the presence of water traces (Martin et al., 2016). This lack of SI-traceability therefore may affect the quality and credibility of $NH_3$ amount fraction values measured in ambient air by national air quality monitoring networks in Europe as well as the spatio-temporal comparability of data across countries.

In this paper, we describe a developed $NH_3$ reference generator, based on permeation and further dynamic dilution, which can generate $NH_3$ RGMs at low amount fractions (1 – 400 nmol/mol). Moreover, we present the set-up of a calibration
method for $NH_3$ analyzers and the validation of the calibration method through a comparison with METAS (Swiss National Metrology Institute). This new reference generator will contribute to fulfil the lack of traceability, improving thus the accuracy and comparability of $NH_3$ datasets.

## 2 Technical choices for the development of the $NH_3$ reference generator

The European MetNH3 project (2014-2017) within the framework of the European Metrology Research Programme
(EMRP) of the European Association of National Metrology Institutes (EURAMET) assessed metrology issues for $NH_3$ in ambient air. The project aimed to improve comparability and reliability of $NH_3$ measurements in ambient air by developing





metrological standards including both reference gas mixtures and standard analytical methods. The target of this development was to provide traceable NH₃ amount fractions in the environmentally relevant range of 0 – 500 nmol/mol (Pogány et al., 2016). Within the project, two solutions were investigated to generate NH₃ amount fractions from 1 to 400

nmol/mol. One solution was based on the production of gravimetric RGMs in gas cylinders at an amount fraction of 10 µmol/mol followed by a dynamic dilution using mass flow controllers. The other one was based on gas permeation with a double dynamic dilution stage (METAS, 2017).

The gravimetric method for preparing RGMs consists of introducing the pure constituent compounds of the RGMs into a cylinder under vacuum, and determining then the quantities of each constituent by weight differences (ISO6142-1, 2015).

Thus, this method is traceable to the international system by kg and allows the assessment of uncertainties of amount fractions. The problem linked to this method is to obtain stable gas mixtures over time. Indeed, NH₃ is a compound that is very easily adsorbed on the internal walls of the cylinder, as shown in the study carried out during the MetNH3 project on different materials (Pogány et al., 2016). This study shows Teflon-perfluoroalkoxy (Teflon-PFA) or SilcoNert® 2000 materials lead to the lowest NH₃ adsorption. As part of the same project, the Dutch National Metrology Institute (VSL)

carried out a stability study on NH₃ gas mixtures at 10 µmol/mol in cylinders treated by different processes. The obtained results showed that the stability of the amount fraction of the gravimetrically prepared gas mixtures at 10 µmol/mol of NH₃ was not completely satisfactory, especially since it was subsequently necessary to dilute this gas mixture dynamically to obtain the desired low amount fractions (nmol/mol).

The principle of the gas permeation technique (ISO6145-10, 2008) is based on the permeation of a pure chemical

compound through a semi-permeable membrane. This technique is particularly used for the generation of dynamic RGMs of gaseous compounds that are reactive or not very stable over time (e.g. sulfur dioxide (SO₂), nitrogen dioxide (NO₂)). During the MetNH3 project, METAS selected the permeation technique to generate NH₃ RGMs. The developed portable device was based on the implementation of a permeation tube in a temperature-regulated oven and on mass flow controllers allowing to dilute dynamically the pure NH₃ from the permeation tube to the desired amount fractions. All parts in contact with NH₃

were inert due to a SilcoNert® 2000 coating to avoid adsorption effects (Vaittinen et al., 2018). The results obtained by METAS during the study showed a very good stability in the generation of NH₃ with short response times, even at very low amount fractions.

The project results showed that it is therefore possible to generate low amount fractions of NH₃ by limiting the adsorption of this compound with coatings such as SilcoNert® 2000. The use of gas permeation provides an NH₃ amount fraction range

compatible with measurements in ambient air with relative uncertainties of less than 3%. The French NMI LNE has therefore chosen this technique for developing the NH₃ reference generator, used for calibrating analyzers over their analytical range. This technique has been used by LNE for many years for molecules such as sulfur dioxide, nitrogen dioxide or even more recently for traces of water vapor which are not stable in gas cylinders at these low amount fractions.



## 3 Development of the NH₃ reference generator

### 3.1 Principle of the permeation method

Permeation tubes consist of a polymeric membrane (usually polytetrafluoroethylene (PTFE) or a copolymer of tetrafluoroethylene and hexafluoropropylene (FEP)) through which a small amount of pure analyte ($NH_3$) will permeate (Lucero, 1971). The most common type is a small, sealed tubular device, which contains the analyte (e.g. $NH_3$) in liquid form. A very small but stable flow (hundreds of ng/min) of ammonia in gaseous phase is emitted through the permeable membrane when the temperature is constant. This emission or permeation rate depends on the physical characteristics of the permeable membrane, on the permeability of the analyte through the membrane, on the temperature of the permeation tube and on the partial pressure across the membrane. The permeation rate of the tube is determined by measuring its mass loss over time using a microbalance under controlled temperature conditions to guarantee the traceability to the SI. The tube placed in a temperature- and flow-controlled stream of inert dilution gas (see Fig. 1) allows to generate an ammonia gas mixture, which is stable in terms of amount fraction at trace levels (nmol/mol).

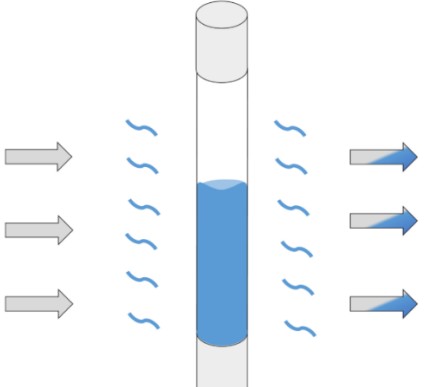

**Figure 1: Principle of the permeation technique. The permeation tube contents the selected pure compound in liquid form (blue). The compound (e.g. $NH_3$) permeates through the membrane (blue wave shapes) at specific permeation rates ($q_m$), which are mainly temperature dependent. Grey arrows represent the dilution gas (e.g. nitrogen, synthetic air) used to generate reference gas mixtures at different amount fractions depending on the flow ($q_v$). The bicolor arrows represents the generated reference gas mixture.**

### 3.2 Description of the NH₃ reference generator

LNE drew up specifications for the development of a NH₃ reference generator based in particular on the NF EN ISO 6145-10 standard (ISO, 2008). The reference generator was implemented by the company 2M PROCESS on the specific request of LNE, based on a very close collaboration involving metrological, technical and software development skills. This generator consists of a thermostatically controlled oven (model 150 Dynacalibrator; VICI Metronics, WA, USA) that houses the NH₃ permeation tube, four mass flow controllers (MFCs) (EL-Flow prestige; Bronkhorst, Netherlands), one back pressure regulator (EL-Press; Bronkhorst, Netherlands) and a touch-sensitive graphic interface for automatic control of all the elements (see Fig. 2).



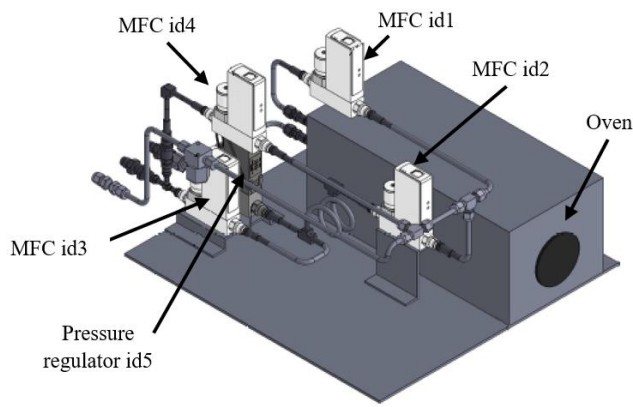


**Figure 2: Scheme of the reference generator implemented by the company 2M PROCESS under LNE's specifications (height: 43 cm; length: 52 cm; width: 60 cm). Labels correspond to the main elements of the reference generator: oven, pressure regulator and mass flow controllers (MFC).**

All parts (tubings, oven and MFCs) in contact with $NH_3$ are coated with SilcoNert® 2000 in order to limit the adsorption

of $NH_3$ on the walls as much as possible. The ammonia permeation tube (99.98 % purity; Fine Metrology S.r.l.s., Italy) is placed inside the temperature-controlled oven at $30.00°C \pm 0.01°C$, which is flushed with purified dry air (on-line gatekeeper CE2500KF04R; ENTEGRIS, USA) at 100 ml/min (id1). The first dilution stage consists of two MFCs (id1 and id2) with a measurement range of respectively 120 ml/min and 10 l/min allowing the generation of amount fractions from 400 to 60 nmol/mol (also depending on the permeation rate of the tube) with a back pressure regulator (1 bar relative) (id5) for

regulating the gas pressure in the permeation oven. The second dilution stage is composed of two MFCs (id3 and id4) with a measurement range of respectively 120 ml/min and 10 l/min leading to amount fractions between 50 and 1 nmol/mol. All MFCs are calibrated against very accurate flowmeters (Molbloc/Molbox™ from Fluke Calibration, WA, USA) previously calibrated by the dynamic gravimetric method at LNE. An integrated software (graphical interface), which allows automatic control of MFCs, calculates the amount fraction of the gas generated as well as its associated uncertainty. Figure 3 shows the

different elements of the reference generator and their functions during the generation of $NH_3$ at the first and second stages.




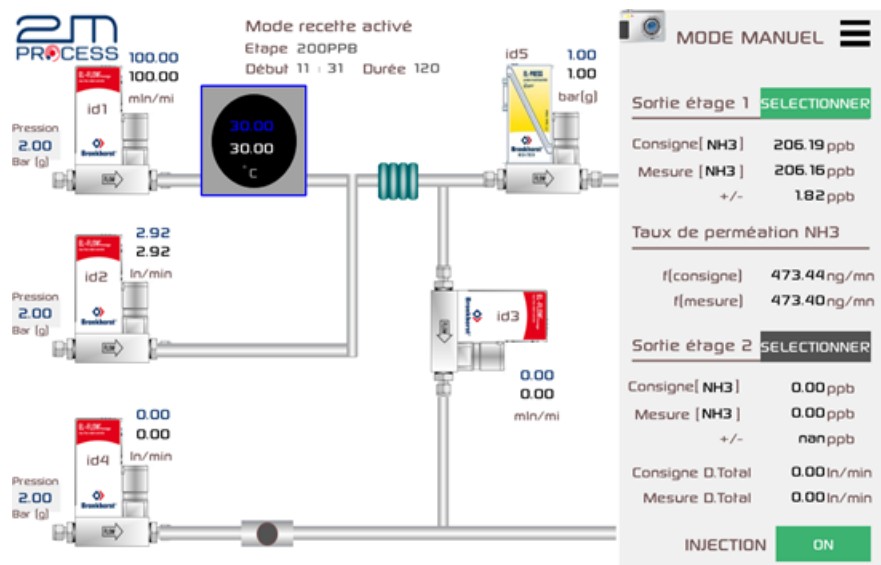

**Figure 3: Schematic diagram of the reference generator. The mass flow controllers (MFCs) used to regulate the carrier and dilution gas are indicated as id1-id4. Id5 represents the pressure controller and the blue square the oven where the permeation**
**tube is placed. On the right side of the figure, an example of set and reading values provided by the software is shown (graphical user interface in French).**

The ammonia amount fraction $x_{NH3}$ generated by the reference generator, in nmol/mol, can be estimated according Eq. (1):

$$x_{NH_3} = \frac{\frac{q_m}{M}}{\frac{qv}{Vn}}$$
(1)

Here, $q_m$ is the permeation rate of the tube in ng/min, $M$ is the molar mass of ammonia in g/mol, $qv$ is the total air flow in l/min and $Vn$ is the normal molar volume of the dilution gas in l/mol (273.15 K and 1.013 bar).

## 4 Calculation of the permeation rate of ammonia tube and the associated uncertainty

The nominal permeation rate (provided by Fine Metrology) of the ammonia tube was (462 ng/min ± 5 %) at 30 °C. The tube was weighed monthly to account for effects of temperature variation and temporal stability on the
determination of the amount of pure $NH_3$ emitted (i.e. permeation rate).

### 4.1 Determination of the permeation rate

The rate of permeation $q_{m_{i-j}}$ (in ng/min) of the tube used in this study between day i and day j was calculated according to Eq. (2):

$$q_{m_{i-j}} = P_{NH3} \times \frac{\Delta m * 10^9}{\Delta t} + C_T + C_R + Sq_{m_{i-j}}$$
(2)





where,

| | |
|---|---|
| $P_{NH3}$ | purity of the liquid NH₃ in the permeation tube |
| $\Delta m$ | mass loss of the tube between day i and day j (in g) |
| $\Delta t$ | period of time for the mass loss $\Delta m$ between day i and day j (in min) |
| $C_T$ | correction accounting for the stability of the temperature in the oven (in ng/min) |
| $C_R$ | correction associated to the reproducibility of the weighing procedure (in ng/min) |
| $Sq_{m_{i-j}}$ | stability of the permeation rate over time (in ng/min) |


## 4.2 Contributors to the uncertainty of the permeation rate

### 4.2.1 NH₃ purity ($P_{NH3}$)

The pure compound in liquid form contained in the permeation tube was supplied by the company Sigma Aldrich. The certificate of analysis provided by the manufacturer stated a purity greater than 99.98 %, being water vapor as the major impurity. The purity was considered to be equal to 0.9999 with a standard uncertainty calculated according Eq. (3) (uniform law):

$$u(P_{NH3}) = \frac{0.0001}{\sqrt{3}} \tag{3}$$

### 4.2.2 Mass loss of the tube ($\Delta m$)

The calculation of the standard uncertainty of the tube mass loss between day i and day j took into account the following uncertainty sources: mass standards, balance resolution and accuracy, weighing repeatability, air density and volume of the tube.

### 4.2.3 Period of time for the mass loss ($\Delta t$)

The standard uncertainty of the period during which the mass loss was determined was calculated by applying a uniform law given by Eq. (4):

$$u(\Delta t) = \frac{2}{\sqrt{3}} \tag{4}$$

### 4.2.4 Stability of temperature ($C_T$)

The correction accounting for the stability of the temperature in the oven was considered as equal to zero. However, an
uncertainty is associated with it, which was calculated using a range of the variations of the oven temperature equal to 0.05 °C. As the influence of temperature on the permeation rate of the tube is of the order of 8% per degree, considering a uniform law (Eq. (5)), the standard uncertainty was:





$$u(C_T) = \frac{0.08 \times 0.05}{\sqrt{3}} \times q_{m_{i-j}} \tag{5}$$

**4.2.5 Reproducibility of the weighing procedure ($C_R$)**

The reproducibility of the tube weighing process was estimated by weighing periodically a permeation tube with a stable permeation rate during 3 months. The reproducibility correction was considered as zero and the associated uncertainty was evaluated as being equal to 0.2 % of the permeation rate. The reproducibility standard uncertainty was calculated according to Eq. (6):

$$u(C_R) = 0.002 \times q_{m_{i-j}} \tag{6}$$

**4.2.6 Evolution of the permeation rate over time ($Sq_{m_{i-j}}$)**

The permeation tube was placed in the oven of the reference generator at a temperature of 30.00 °C ± 0.05 °C; it was weighed regularly to calculate its permeation rate over a period of two years. Table 1 summarizes the different determinations of the permeation rate of the tube (using Eq. (2)) as well as the associated uncertainties.

**Table 1: Estimated permeation rate over time and its uncertainty; k is the coverage factor of the uncertainty.**

| Date | Permeation rate ($q_{m_{i-j}}$) (ng/min) | Expanded uncertainty (k=2) (ng/min) |
|---|---|---|
| 10/05/2019 | 473.4 | 3.1 |
| 14/06/2019 | 475.3 | 3.1 |
| 19/07/2019 | 486.8 | 3.2 |
| 02/09/2019 | 487.9 | 3.1 |
| 27/09/2019 | 482.5 | 3.3 |
| 08/11/2019 | 475.3 | 3.0 |
| 13/12/2019 | 475.4 | 3.1 |
| 16/01/2020 | 470.8 | 3.1 |
| 24/02/2020 | 476.2 | 3.1 |
| 11/05/2020 | 478.4 | 3.0 |
| 12/06/2020 | 473.4 | 3.1 |
| 17/07/2020 | 480.3 | 3.1 |
| 21/08/2020 | 475.4 | 3.1 |
| 25/09/2020 | 473.1 | 3.1 |
| 29/10/2020 | 474.7 | 3.1 |
| 04/12/2020 | 477.2 | 3.1 |
| 08/01/2021 | 474.8 | 3.1 |
| 12/02/2021 | 474.6 | 3.1 |
| 19/03/2021 | 474.4 | 3.1 |



The results obtained over two years showed that the permeation rate of the NH$_3$ tube was stable over time with the exception of the first few months, which is quite usual when putting a new permeation tube into an oven ("tube maturation"). Figure 4 illustrates the change in the permeation rate of the tube over time. The error bars correspond to the expanded uncertainty (k = 2) on the determination of the permeation rate of the tube.

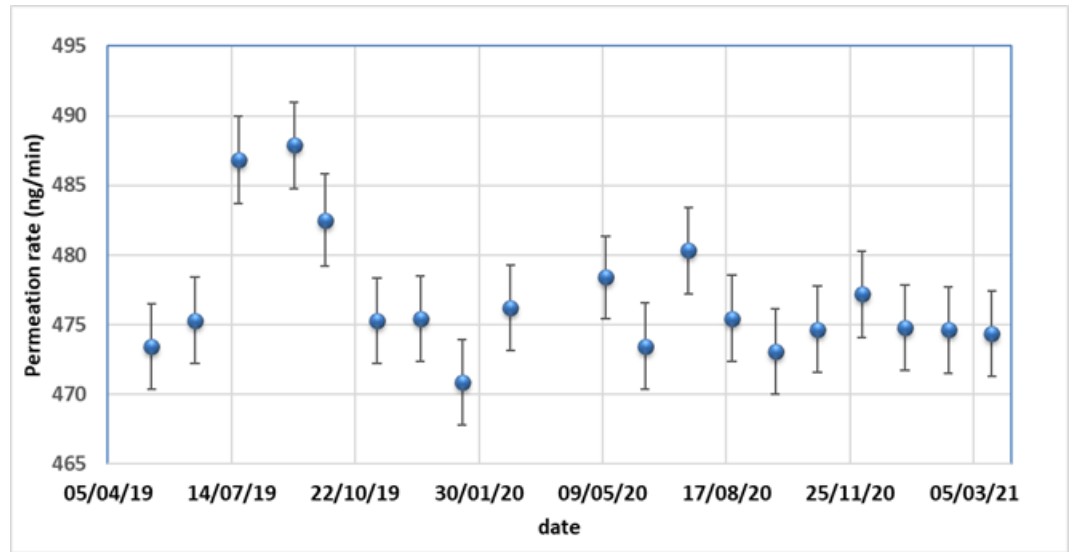

**Figure 4: Evolution of the permeation rate of the tube over time; the points represent the different values of the permeation rate measured over time; the error bars correspond to the expanded uncertainty (k = 2) on the determination of the tube permeation rate.**

The stability over time of the permeation rate ($Sq_{m_{i-j}}$) was considered as equal to zero. Its standard uncertainty $u(Sq_{m_{i-j}})$, in ng/min, was calculated by considering the maximum deviation found between the last three determinations of the tube permeation rate (Eq. (7)):

$$u(Sq_{m_{i-j}}) = \frac{Maximum\ deviation}{2\sqrt{3}} \qquad (7)$$

### 4.3 Uncertainty budget of the NH₃ permeation rate

The uncertainty budget on the NH$_3$ permeation rate was calculated using the standard uncertainties determined previously and by applying the law of propagation of uncertainties expressed in the Guide to the expression of uncertainty in measurement (GUM) (JCGM, 2008) on the equation 2. One example is summarized in Table 2.



**Table 2: Uncertainty budget of one of the permeation rates used in this study. The contribution, in percentage, of each variable to the overall uncertainty is indicated in the column Weight. The last row shows the permeation rate value and its expanded uncertainty; k is the coverage factor of the uncertainty.**

| Variable $X_i$ | Variable name | Units | Value | Standard uncertainty $u(X_i)$ | Weight (%) |
|---|---|---|---|---|---|
| $P_{NH3}$ | NH₃ purity of the tube | - | 0.9999 | 5.77E-05 | 0.03 |
| $\Delta m$ | Mass loss of the tube | g | 2.3286E-02 | 2.66E-05 | 10.58 |
| $\Delta t$ | Period of time for the mass loss | min | 49057 | 1.15E+00 | 0.00 |
| $C_t$ | Correction due to the influence of the variations of temperature | ng/min | 0 | 1.10E+00 | 43.14 |
| $C_R$ | Reproducibility of the permeation rate determination | ng/min | 0 | 9.00E-01 | 29.04 |
| $Sq_{m_{i-j}}$ | Evolution of the permeation rate over time | ng/min | 0 | 6.93E-01 | 17.21 |

255

| $q_{m_{i-j}}$ | 474.6 ± 3.4 nmol/mol (k=2) |
|---|---|

## 5 Calculation of the NH₃ amount fraction in the generated RGMs and the associated uncertainty

### 5.1 Determination of the amount fraction of the generated RGMs

The developed reference generator has two dilution stages, the first one for "higher" amount fractions (400 nmol/mol to 60
260 nmol/mol) and the second one for lower amount fractions (50 nmol/mol to 1 nmol/mol). NH₃ amount fractions in the RMGs generated by the reference generator can be estimated applying Eq. (8) and Eq. (9):

- After the first stage of generation

$$x_{ref1} = 10^{-9} \times \left[ \frac{q_{m_{i-j}}/M}{(qva+qvb)/V_n} + Imp \right] \tag{8}$$

265

- After the second stage of generation





$$x_{ref2} = 10^{-9} \times \frac{qvc \times \left(\frac{q_{m_{i-j}}}{M \times \left(\frac{qvb+qva}{Vn}+\frac{q_{m_{i-j}}}{M}\right)}+Imp\right)}{qvd+qvc} \tag{9}$$

where,

| | |
|---|---|
| $x_{ref1}$ | NH$_3$ amount fraction in the RGMs after the first stage of generation (nmol/mol) |
| $x_{ref2}$ | NH$_3$ amount fraction in the RGMs after the second stage of generation (nmol/mol) |
| $q_{m_{i-j}}$ | calculated permeation rate (g/min) |
| $M$ | NH$_3$ molar mass (g/mol) |
| $q_{va}$ | air flow flushing the permeation tube (l/min) |
| $q_{vb}$ | first stage dilution air flow (l/min) |
| $q_{vc}$ | flow rate of the NH$_3$ reference gas mixture from the first stage (l/min) |
| $q_{vd}$ | second stage dilution air flow (l/min) |
| $Vn$ | normal molar volume of the dilution gas (l/mol) |
| $Imp$ | specification of the residual amount fraction after the filter given by the supplier (mol/mol) |

## 5.2 Determination of the standard uncertainties of the amount fractions of the generated RGMs

The variance on the NH$_3$ amount fraction in the generated RGMs was determined using the law of propagation of uncertainty expressed in the Guide to the expression of uncertainty in measurement (GUM) (JCGM, 2008). Standard uncertainties of the NH$_3$ amount fraction of the generated RGMs were as follows:

- Standard uncertainty of the permeation rate $\left(u(q_{m_{i-j}})\right)$ was calculated as described in section 4.3

- Standard uncertainty of the NH$_3$ molar mass ($u(M)$) was determined with data from the IUPAC document (IUPAC, 2013)

- Standard uncertainties of the flow rates ($u(q_{va})$, $u(q_{vb})$, $u(q_{vc})$ and $u(q_{vd})$) were determined using the results obtained during the calibration of the MFCs at LNE. These standard uncertainties take into account the uncertainty of the linearity ($u(linearity)$, which was estimated according a model WLS RegPoly developed by LNE (Yardin, 2013), the uncertainty related to the calibration ($u(calibration)$) and the uncertainty related to the durability of the calibration ($u(durability)$) as shown in Eq. (10):

$$u(qv) = \sqrt{u(linearity)^2 + u(calibration)^2 + u(durability)^2} \tag{10}$$





- Standard uncertainty of the normal molar volume $(u(Vn))$, which is given in the CODATA document (Mohr, 2014)

- Standard uncertainty of the residual $NH_3$ amount fraction in the dilution air $(u(Imp)$. According to the specifications of the dilution gas (Alphagaz 2 from Air Liquide), the residual $NH_3$ amount fraction was less than 10 pmol/mol. Because the dilution gas was further purified by means of an Entegris filter, the residual amount fraction was considered to be equal to 0.005 nmol/mol. The standard uncertainty was estimated following Eq. (11) (uniform law):

$$u(Imp) = \frac{0,005}{\sqrt{3}} = 0,003 \; nmol/mol \tag{11}$$

### 5.3 Summary of the uncertainty budget of the amount fractions of the generated reference gas mixtures

The uncertainty budget of the $NH_3$ amount fractions generated by the reference generator was calculated from the standard uncertainties determined previously and by applying the law of propagation of uncertainties. Two examples at the extreme points of the generation range of the reference generator are summarized in Tables 3 and 4.

**Table 3: Uncertainty budget of the amount fraction of one of the ammonia RGMs generated after the first dilution stage. The contribution, in percentage, of each variable to the overall uncertainty is indicated in the column Weight. The last row shows the amount fraction value and its expanded uncertainty; k is the coverage factor of the uncertainty.**

| Variable $X_i$ | Variable name | Units | Value | Standard uncertainty $u(X_i)$ | Weight (%) |
|---|---|---|---|---|---|
| $q_{m_{i-j}}$ | Permeation rate | g/min | $475.5 \; 10^{-9}$ | $1.65 \; 10^{-9}$ | 46.15 |
| M | $NH_3$ molar mass | g/mol | 17.0305 | $4.1 \; 10^{-4}$ | 0 |
| qva | MFC (id1) | l/min | 0.100002 | $3.0 \; 10^{-4}$ | 0.15 |
| qvb | MFC (id2) | l/min | 1.4601 | $5.8 \; 10^{-3}$ | 53.7 |
| $V_n$ | Molar volume | l/mol | 22.414 | $1.3 \; 10^{-5}$ | 0 |
| Imp | $NH_3$ residual | mol/mol | $5 \; 10^{-12}$ | $3 \; 10^{-12}$ | 0 |

| | | | |
|---|---|---|---|
| **Xref1** | $401.1 \pm 4.2$ nmol/mol (k=2) | | |





**Table 4: Uncertainty budget of the amount fraction of one of the ammonia RGMs generated after the second dilution stage. The contribution, in percentage, of each variable to the overall uncertainty is indicated in the column Weight. The last row shows the amount fraction value and its expanded uncertainty; k is the coverage factor of the uncertainty.**

| Variable $X_i$ | Variable name | Units | Value | Standard uncertainty $u(X_i)$ | Weight (%) |
|---|---|---|---|---|---|
| $q_{m_{i-j}}$ | Permeation rate | g/min | $475.5\ 10^{-9}$ | $1.65\ 10^{-9}$ | 20.65 |
| $M$ | NH$_3$ molar mass | g/mol | 17.0305 | $4.1\ 10^{-4}$ | 0 |
| $qva$ | MFC (id1) | l/min | 0.1000 | $3.0\ 10^{-4}$ | 0 |
| $qvb$ | MFC (id2) | l/min | 6.100 | 0.025 | 26.5 |
| $qvc$ | MFC (id3) | l/min | 0.05999 | 0.00018 | 15.1 |
| $qvd$ | MFC (id4) | l/min | 5.000 | 0.020 | 26.75 |
| $Vn$ | Molar volume | l/mol | 22.414 | $1.3\ 10^{-5}$ | 0 |
| $Imp$ | NH$_3$ residual | mol/mol | $5\ 10^{-12}$ | $3\ 10^{-12}$ | 11.0 |

| $X_{ref2}$ | $1.196 \pm 0.019$ nmol/mol (k=2) |
|---|---|

These results highlighted that the main source of uncertainty of the NH$_3$ reference amount fractions generated after the single dilution (first stage from 400 to 60 nmol/mol) was linked to the calculation of the permeation rate of the tube and to the measurement of the air dilution flow rate (MFC – id2). After the second dilution step (< 60 nmol/mol), the weight of the uncertainties was more evenly distributed among the uncertainty of the permeation rate, the flow rates (MFCs – id2, id3 and id4) and the residual NH$_3$ impurities in the dilution air. The expanded relative uncertainties of the generated amount fractions were between 1 and 2 % over the whole range.

## 6 Characterization of the calibration method of analyzers

### 6.1. Calibration procedure of NH$_3$ analyzers

The reference generator was permanently switched on during the characterization and the NH$_3$ permeation tube was continuously flushed by a synthetic air flow at 100 ml/min. NH$_3$ amount fractions were generated during 2 hours for each amount fraction level using the reference generator. The average of the generated amount fractions and the average of the readings of the analyzer to be calibrated were calculated over the last 30 minutes along with their standard deviations.





## 6.2 Determination of the repeatability of the calibration method

To evaluate the repeatability of the calibration method, nine amount fraction levels were generated in the range between 1 and 400 nmol/mol. The software of the developed reference generator calculated, every ten seconds, the amount fraction of

NH$_3$ generated according to the set parameters (permeation rate, molar mass, purity, etc.), and the measured flow rates (MFCs) and oven temperature (taking into account temperature variations). Table 5 shows the standard deviations calculated over the last 30 minutes of generation for the amount fractions range between 1 nmol/mol and 400 nmol/mol.

**Table 5: Repeatability of amount fractions generated by the ammonia reference generator. The average and the absolute and relative standard deviations of the amount fractions generated during the last 30 minutes are included.**

| Average amount fraction (nmol/mol) | Absolute standard deviation (nmol/mol) | Relative standard deviation (%) |
|---|---|---|
| 400.69 | 0.29 | 0.07 |
| 300.53 | 0.24 | 0.08 |
| 200.38 | 0.14 | 0.07 |
| 100.824 | 0.072 | 0.07 |
| 50.404 | 0.038 | 0.08 |
| 30.434 | 0.024 | 0.08 |
| 10.2142 | 0.0073 | 0.07 |
| 5.1033 | 0.0037 | 0.07 |
| 1.21360 | 0.00078 | 0.06 |


The relative standard deviation of the generated amount fractions at all generation levels was less than 0.1 %, indicating a very good stability of the flow rates and a good temperature control. A standard uncertainty of the repeatability equal to 0.1 % (id. $u(r) = 0.001 \times x_{ref}$) was retained and combined with the uncertainty calculated for the reference generator (cf.

paragraph 5.3).

## 6.3 Determination of the reproducibility of the calibration method

LNE used METAS ammonia analyzer, based on cavity ring-down spectroscopy (G2103, Picarro Inc., CA, USA), for a bilateral comparison. This device is specifically manufactured for the measurement of NH$_3$. Moreover, the analyzer was

upgraded in 2019, when the cavity and the sampling handling were replaced by SilcoNert® 2000 coated elements, thus





limiting adsorptions. The analyzer required an inlet flow rate of 1.8 l/min, which was provided by an external diaphragm vacuum pump (N920AP.29.18, KNF Neuberger Inc., NJ, USA) and was calibrated without particles filter. The tests were performed using Air Alphagaz2 (Air Liquide) purified with ENTEGRIS filter and 1/4 in. coated (SilcoNert® 2000) stainless steel tubing. In order to quantify the reproducibility standard deviation on the calibration of an $NH_3$ analyzer, the calibration

of the Picarro G2103 analyzer was performed 4 times on different days between 20 and 125 nmol/mol (amount fractions are limited by the sampling flow rate of the analyzer). The calibration function was determined for each of the calibrations using the RegPoly software developed at LNE. This software allowed the integration of uncertainties of the X and the Y axis. The procedure was the same as when determining repeatability: generation during 2 hours and calculation of the amount fractions average and associated standard deviations over the last 30 minutes. Tables 6 and 7 summarize the results obtained.


**Table 6: Tests results performed to determine reproducibility of the calibration of an ammonia analyzer using the reference generator from LNE and the Picarro $NH_3$ analyzer from METAS. The coverage factor (k) is indicated for the combined uncertainty of the permeation. The calibration function is also included, where Y are the readings, x are the average $NH_3$ amount fractions generated, b is the slope and $b_0$ the intercept. (\*) This value corresponds to the specification of the residual amount**
**fraction of ENTEGRIS filter.**

| | LNE's reference generator/Picarro G2103 (METAS) | | | | | | |
|---|---|---|---|---|---|---|---|
| | LNE's reference generator (nmol/mol) | | | Picarro G103 (nmol/mol) | | Calibration function $Y = b \cdot x + b_0$ | |
| Date | Average $NH_3$ amount fraction generated (last 30 min) | Standard deviation | Permeation combined uncertainty (k=1) | Average of the readings (last 30 min) | Standard deviation of the readings | $b_0$ (nmol/mol) | b |
| 15/05/2020 | 20.36 | 0.013 | 0.18 | 21.47 | 0.20 | 0.3193 | 1.0318 |
| | 77.73 | 0.054 | 0.54 | 80.23 | 0.29 | | |
| | 124.68 | 0.089 | 0.86 | 129.14 | 0.43 | | |
| | 0.005* | | 0.003 | 0.31 | 0.35 | | |
| 19/05/2020 | 20.36 | 0.015 | 0.17 | 21.62 | 0.20 | 0.4260 | 1.0321 |
| | 77.74 | 0.057 | 0.54 | 80.45 | 0.29 | | |
| | 124.67 | 0.084 | 0.85 | 129.21 | 0.43 | | |
| | 0.005* | | 0.003 | 0.35 | 0.35 | | |
| 25/05/2020 | 20.36 | 0.013 | 0.17 | 21.70 | 0.21 | 0.4510 | 1.0320 |
| | 77.74 | 0.052 | 0.54 | 80.48 | 0.30 | | |
| | 124.68 | 0.086 | 0.86 | 129.22 | 0.44 | | |
| | 0.005* | | 0.003 | 0.32 | 0.35 | | |
| 27/5/2020 | 20.36 | 0.016 | 0.17 | 21.62 | 0.21 | 0.3734 | 1.0330 |
| | 77.73 | 0.052 | 0.54 | 80.50 | 0.30 | | |
| | 124.67 | 0.089 | 0.86 | 129.24 | 0.44 | | |
| | 0.005* | | 0.003 | 0.25 | 0.38 | | |





The reproducibility of the determination of the calibration function was around 0.05 % (Table 7). A reproducibility uncertainty increased to 0.1 % (id. $u(repro) = 0.001 \times x_{ref}$) was considered in the uncertainty budget.

**Table 7: Reproducibility of the determination of the calibration function of an ammonia analyzer**

|  | Slope | Intercept (nmol/mol) |
|---|---|---|
| 15/5/2020 | 1.0318 | 0.3193 |
| 19/5/2020 | 1.0321 | 0.4260 |
| 25/5/2020 | 1.0320 | 0.4510 |
| 27/5/2020 | 1.0330 | 0.3734 |
| **Average** | 1.03223 | 0.392 |
| **Standard deviation** | 0.00053 | 0.059 |


**6.4 Uncertainty budget of the calibrated amount fractions**

The expanded uncertainty (k = 2) of the calibrated $NH_3$ amount fractions ($U_{NH3}$) in nmol/mol was calculated using Eq. (12):

$$U_{NH3} = 2 * \sqrt{u^2(x_{ref}) + u^2(r) + u^2(repro)} \qquad (12)$$


where,

$u^2(F)$      variance of the $NH_3$ amount fraction calculated by the software of the reference generator (in nmol/mol)

$u^2(r)$      repeatability variance of the calibration method in nmol/mol

$u^2(repro)$ reproducibility variance of the calibration method in nmol/mol


Table 8 shows the final uncertainties obtained over the range of amount fractions generated by the reference generator.

**Table 8: Uncertainty budget of the $NH_3$ amount fractions used as calibration values. The coverage factor (k) is indicated for the absolute and the relative expanded uncertainties.**

| $NH_3$ reference amount fractions (nmol/mol) | Absolute expanded uncertainty (nmol/mol) ($U_{NH3}$) (k=2) | Relative expanded uncertainty (%) ($U_{NH3}$) (k=2) |
|---|---|---|
| 400.56 | 4.36 | 1.1 |
| 297.56 | 3.27 | 1.1 |
| 201.58 | 2.23 | 1.1 |



| 100.79 | 1.12 | 1.1 |
| 50.07 | 0.70 | 1.4 |
| 25.15 | 0.36 | 1.4 |
| 9.996 | 0.15 | 1.5 |
| 4.998 | 0.07 | 1.5 |
| 1.1951 | 0.02 | 1.6 |


The uncertainty budget of the $NH_3$ reference amount fractions highlighted that over an amount fraction range between 1 and 400 nmol/mol, the relative expanded uncertainties was between 1.1 % and 1.6 %. This uncertainty may vary slightly over time depending mainly on the uncertainties of the permeation rate of the $NH_3$ tube and the MFC calibrations. Nevertheless, taking into account the experience acquired at LNE on the weighing of permeation tubes and on the calibration

of MFCs, the relative expanded uncertainties would remain less than 2 % (k = 2) over the whole range studied. It should be noted that for the amount fractions obtained after the first dilution stage, the associated uncertainty was mainly linked to the uncertainty of the permeation rate and the dilution flow rate. For the amount fractions generated after the second dilution stage, the main uncertainties were those of the $NH_3$ tube permeation rate, the two dilution flow rates (id3 and id4) and the flow rate taken from the first stage (id2). The uncertainty of the amount fraction of the $NH_3$ impurities present in the dilution

gas could represent up to 10 % of the final uncertainty for very low amount fractions.

Figure 5 shows schematically the distribution of the different uncertainty sources for $NH_3$ calibrated amount fractions from 1 nmol/mol (second stage) and 400 nmol/mol (first stage).

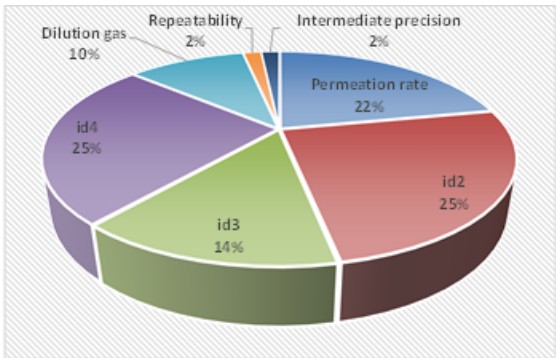
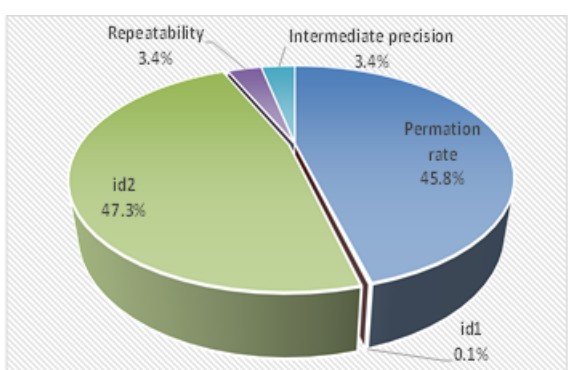

**Figure 5: Distribution of the different uncertainty sources for $NH_3$ calibrated amount fractions from 1 nmol/mol (second stage of dilution; left) and 400 nmol/mol (first stage of dilution; right). The sources indicated as id1-id4 correspond to the mass flow controllers of the generator.**





**7 Validation of the calibration method by a bilateral comparison with METAS**

A bilateral comparison with METAS was organized in order to verify the accuracy of the calibration method developed at
LNE. In March 2020, METAS calibrated their PICARRO G2103 gas analyzer using a $NH_3$ permeation tube (99.98 % purity;
Fine Metrology S.r.l.s., Italy). The permeation rate of the tube was determined at METAS using a coated (SilcoNert® 2000)
magnetic suspension balance (MSB; Rubotherm GmbH, Germany) under regulated temperature and pressure conditions. The
permeation rate value was 206.6 ng/min with an expanded uncertainty (U) of 3.1 ng/min (k = 2) at (37.96 ± 0.01) °C.
METAS calibrated their instrument on 02/03/2020. For that purpose, METAS used the dilution system of their MSB, to
generate reference gas mixtures of $NH_3$ in synthetic air (99.9996 % purity; PanGas AG, Switzerland). Three calibration
points were generated at amount fractions between 35 and 150 nmol/mol. Measurements of synthetic air were used as zero
calibration point. Each calibration point was generated during 90 minutes; only the average value of the last 30 minutes of
generation and reading were considered (Table 9). The tubing used during the generation was 1/4 in. coated (SilcoNert®
2000) stainless steel.

**Table 9: Results of the Picarro G2103 analyzer calibration performed by METAS. The generated amount fraction and reading
values are the average of the last 30 minutes of generation and reading, respectively. The coverage factor (k) of the expanded
uncertainty of the generation is indicated.**

| Generated reference value (nmol/mol) | | Picarro G2103 analyzer (nmol/mol) | |
| --- | --- | --- | --- |
| Generated NH₃ amount fractions | Absolute expanded uncertainty (k=2) | Readings | Standard deviation of the readings |
| 0.025 | 0.029 | 1.06 | 0.23 |
| 37.32 | 0.66 | 38.82 | 0.23 |
| 98.61 | 1.69 | 101.73 | 0.36 |
| 147.23 | 2.52 | 155.71 | 0.59 |

The analyzer's calibration function estimated using the RegPoly software developed by LNE followed Eq. (13):

$$Y = 1.046 \cdot X + 0.22 \tag{13}$$

where:

$Y$      reading of the analyzer

$X$      reference amount fraction

METAS' analyzer was then calibrated four times by LNE at the end of May 2020 according to the developed method.
The linear regression of each calibration was calculated from RegPoly software. The uncertainties of X and Y were taken



into account. The average and standard deviation of the four slopes (b) were calculated in order to estimate the expanded uncertainty (U) of the determination of the slope (Eq. (14)).


$$U = k \cdot \sqrt{u(b)^2 + std^2} \qquad (14)$$

where $k$ is the coverage factor (k = 2), $u(b)$ is the uncertainty of the determination of b (RegPoly) and $std$ is the standard deviation of the 4 determinations of b (LNE).

The normalized deviation ($En$) between the slopes of the regression lines determined by METAS and by LNE, was equal to 0.54 (Table 10). This value lower than 1 demonstrated the good agreement between the $NH_3$ reference generators developed by METAS and by LNE. This comparison validated the LNE reference generator and the analyzer's calibration method.

**Table 10: Result comparison between LNE and METAS; b is the slope of the calibration functions; Std is the standard deviation of the calibration slopes obtained by LNE between 15-27 May 2020; u(b) is the standard uncertainty of the slope determination; U(b)**
**is the expanded uncertainty of the slope determination and k its coverage factor. En is the normalized deviation between LNE and METAS slopes.**

|          | LNE    | METAS  |
|----------|--------|--------|
| b        | 1.032  | 1.046  |
| Std      | 0.0005 | -      |
| u(b)     | 0.0091 | 0.0082 |
| U(b) k=2 | 0.019  | 0.017  |

|    | En     |
|----|--------|
|    | **0.54** |

## 8 Implementation of the metrological procedure to calibrate a NH₃ analyzer

Ambient air quality monitoring in France is entrusted to 18 Air Quality Monitoring Networks (AASQA) which perform ammonia concentrations measurements in ambient air with different types of measurement techniques including automatic analyzers. To ensure the reliability of the $NH_3$ measurements performed by these analyzers LNE developed a metrological calibration procedure. The calibration procedure consists of generating dynamic RGMs at different ammonia amount fractions (at least 5 levels in synthetic air) on the measurement range defined by the AASQA (between 1 to 400 nmol/mol)
using the reference generator. These RGMs are then injected in $NH_3$ analyzers and, for each level, the reference amount fraction and the reading are compared. Figure 6 presents the recordings of the ammonia reference amount fractions and the





amount fractions measured by the analyzer over the range from 1 nmol/mol to 300 nmol/mol. The standard deviations over the last 30 minutes were calculated and given in Figure 6.

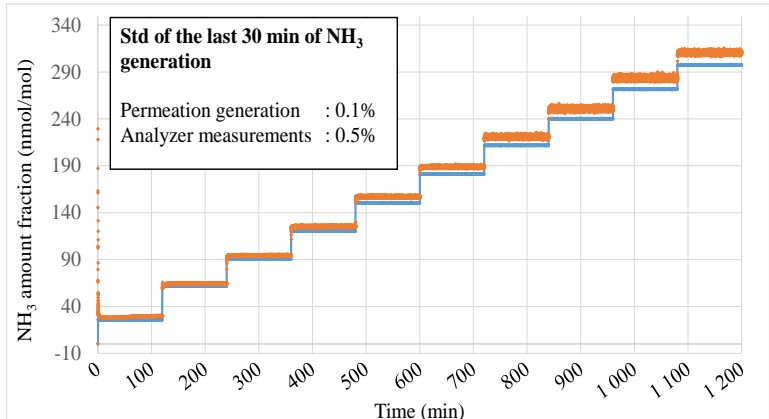

**Figure 6: Stability of the ammonia reference amount fractions (blue line) and of the analyzer readings (orange close circles) over the range from 1 nmol/mol to 300 nmol/mol.**

These results demonstrate a very high stability of the ammonia reference amount fraction calculated from the measurement of the various parameters of the reference generator (MFCs flow rates, oven temperature). It can be noted that the MFC regulation is very fast, as suggested by the immediate response time when changing the setpoint. Furthermore, the results show a high stability of the flow rates over time, because no drift in the calculated amount fraction is observed and its standard deviation is less than 0.1 % for the whole range. This highlights the good quality of the MFCs and oven temperature control. Analysis of the $NH_3$ amount fractions read by the analyzer shows a low standard deviation on the measurements (0.5 %).

Figure 7 shows the calibration line between the averages of the reference amount fractions and the readings of the $NH_3$ analyzer number 2244-AHDS2115 (G2103, Picarro Inc., CA, USA). This calibration line is representative of those obtained during the calibration of all tested analyzers since the end of 2020.





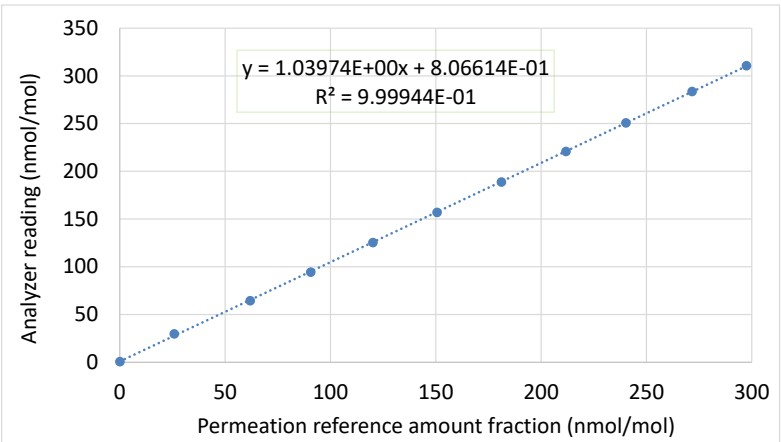

**Figure 7: Typical calibration line obtained for a NH₃ analyzer (dash line). Blue close circles indicate the analyzer response for each generated NH₃ reference (in both cases, average of the last 30 minutes of reading and generation). The error bars corresponding to the standard deviation of the analyzer reading (y-axis) and of the generated references (x-axis) are hidden by the blue close circles.**

By the end of 2020, LNE calibrated seven NH₃ analyzers set up by four French networks in monitoring stations. The calibration results determined by LNE for each analyzer are given in Table 11.

**Table 11: Calibration results (slope, intercept and correlation coefficient) determined by LNE for seven NH₃ analyzers used by four French networks in monitoring stations**

| Trademark | Model | Analyzer identification | Slope | Intercept (nmol/mol) | Correlation coefficient |
|---|---|---|---|---|---|
| Picarro | G2103 | 3711-AHDS 2168 | 1.0357 | 1.1089 | 0.9999 |
| Picarro | G2103 | 3126-AHDS2105 | 0.9869 | 1.4538 | 0.9994 |
| Picarro | G2103 | 3724-AHDS2169 | 1.0386 | 0.8520 | 1.0000 |
| Picarro | G2103 | 2244-AEDS2059 | 0.9890 | 3.4323 | 0.9998 |
| Picarro | G2103 | 2244-AHDS2115 | 1.0397 | 0.8066 | 0.9999 |
| Picarro | G2103 | 2495-AEDS2068 | 1.0020 | 3.0717 | 0.9996 |
| Picarro | G2103 | 2356-AHDS2064 | 0.9901 | 0.9986 | 1.0000 |

The results summarized in Table 11 show that the calibration can be very different from one to another NH₃ analyzer of the same trademark and model up to an accuracy error of 4 %. The calibration process is therefore unavoidable to ensure the reliability of NH₃ measurements performed in ambient air by the monitoring networks to ensure their spatiotemporal comparability. These calibration tests also demonstrated the good linearity of all NH₃ analyzers with correlation coefficients between 0.9994 and 1.000.



## 9 Conclusion

This study led to the development of a reference generator for ammonia in air based on the dynamic generation method by gas permeation over a range of amount fractions from 1 to 400 nmol/mol, in close collaboration with the company 2M PROCESS according to the specifications established by LNE. The reference generator integrates a real-time calculation of the ammonia generated amount fractions (1 to 400 nmol/mol) and its associated expanded uncertainties and allows the calibration of $NH_3$ analyzers in an automatic way. All parameters and measurements carried out are stored in a downloadable file so that they can be easily processed in Excel files. The characterization of this reference generator led to relative expanded uncertainties of the ammonia reference amount fractions in the reference gas mixtures lower than 2 % (k = 2). This result is very satisfactory considering the very low levels of the amount fractions of interest and the potential ammonia adsorption/desorption on the surfaces in contact.

The bilateral comparison organized by LNE and METAS consisted of the calibration of a PICARRO G2103 gas analyzer by both institutes, which used reference gas mixtures generated dynamically by two different reference generators. The normalized deviation between the slopes of the regression lines determined by METAS and by LNE was equal to 0.5, which highlights the good agreement between the dynamic $NH_3$ reference generators developed by METAS and by LNE. This comparison allowed the validation of LNE's reference generator and the analyzer calibration method developed.

Furthermore, the tests carried out during this study showed the importance of the internal surface treatment of calibrated analyzers in order to limit ammonia adsorption strongly.

Since the end of 2020, several French air quality monitoring networks have requested LNE to calibrate their $NH_3$ analyzers, enabling them to guarantee the traceability and accuracy of measurements performed in France.

*Data availability.* Research data could be made available after review of the request.

*Author contribution.* TM and CS conceived the idea, methods and experimental design of the study; MI calibrated, calculated the uncertainty and provided the analyzer for the validation; CS designed the reference generator, carried the experiments out, analysed the data and calculated the uncertainties of the measurements; TM led the writing of the manuscript; CS, SV, MI, CP and BN contributed critically to the drafts. All authors gave final approval for publication.

*Competing interests.* The authors declare that they have no conflict of interest.

*Acknowledgment.* We thank Mr Steve Belliride from Sensor and Mr Marc Monnet, director of 2M PROCESS company for their strong involvement in this study. Owing to their experience and their skills in the field of measuring instruments used for gas analysis, the ammonia reference gas generator was developed with accurate metrological performances in a short time. The research leading to these results was performed within the French National laboratory for monitoring air quality



(LCSQA) program and received funding from the French Ministry of Environment (MTE). Programs under Grant
Agreements MTE no. 2201125723, no. 2201192949 and no. 2201243758.

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
