# Peer review of "Air pollution monitoring: Development of ammonia (NH3) dynamic reference gas mixtures at nmol/mol levels for improving the lack of traceability of measurements"

_Atmospheric Measurement Techniques, 2022_

## Author Response (AR1)

**Replies to Referee #2's comments**

The authors are grateful for the valuable comments and suggestions done by the anonymous referee #1 (in blue). Our replies to the comments and suggestions are given below (in black).

General comments:

This manuscript developed ammonia dynamic reference gas mixtures at nmol/mol levels in the lab. I agree with the authors that such a reference gas could improve the QA/QC in ammonia measurements, which is highly needed in field investigations. The data and techniques used in this study are well described, referenced and easy to follow. The conclusions are consistent with the stated objectives. This study represents an original and interesting contribution to the determination of ammonia behavior in the air. My recommendation is to go through some minor revisions before accepting for publication.

Thanks for the positive feedback.

Major concerns:

This study did not indicate the reasons to select the measurement range of 1-400 nmol/mol in this study (e.g., Line 454). The readers may wonder if it is representative of ambient ammonia concentrations worldwide, and thus the reference gas mixtures can be widely used.

$NH_3$ amount fractions in ambient air, measured at different monitoring stations globally distributed, are lower than 100 nmol/mol (Aksoyoglu et al., 2020; Park et al., 2021; von Bobrutzki et al., 2010). By selecting the range 1 – 400 nmol/mol, we aimed to cover the $NH_3$ amount fractions in ambient air (immissions), as well as the operating range of the analysers used to measure atmospheric $NH_3$.

The reason behind our choice has been added to the text in L17; this information has been also included in the Introduction.

Aksoyoglu, S., Jiang, J., Ciarelli, G., Baltensperger, U., and Prévôt, A.S.H.. Role of ammonia in European air quality with changing land and ship emissions between 1990 and 2030. Atmospheric Chemistry and Physics, 20, 15665 – 15680, https://doi.org/10.5194/acp-20-15665-2020, 2020.

Park, J., Kim, E., Oh, S., Kim, H., Kim, S., Kim Y.P., Song, M. Contributions of ammonia to high concentrations of PM2.5 in an urban area. Atmosphere, 12, 1676. https://doi.org/10.3390/atmos12121676, 2021.

Von Bobrutzki, K., Braban, C. F., Famulari, D., Jones, S. K., Blackall, T., Smith, T. E. L., Blom, M., Coe, H., Gallagher, M., Ghalaieny, M., McGillen, M. R., Percival, C. J., Whitehead, J. D., Ellis, R., Murphy, J., Mohacsi, A., Pogany, A., Junninen, H., Rantanen, S., Sutton, M. A., and Nemitz, E. Field inter-comparison of eleven atmospheric ammonia measurement techniques, Atmos. Meas. Tech., 3, 91–112, https://doi.org/10.5194/amt-3-91-2010, 2010.

Line 83: Could you please change the unit of nmol/mol in the whole text to ppb or ug/m3, which are more common in literature reporting ammonia concentrations?

According to the International System of Units (SI-units), the unit for amount fractions is mol/mol. As this manuscript is related to gas metrology, we are following the SI-units (cf. International vocabulary of metrology – Basic and general concepts and associated terms (VIM) and works made in the framework of BIPM). Nevertheless, we have modified the introduction by adding precisions as follows: "The method is based on gas permeation and further dynamic dilution to obtain an amount fraction range between 1 and 400 nmol/mol (also well known as ppb; 1 ppb ($NH_3$) » to 0.7 µg/m$^3$) to cover the $NH_3$ amount fractions measured in ambient air (immissions) by the monitoring networks."

Why Picarro G2103 analyzer was selected as a reference in this study (e.g., Line 360)?

We planned a bilateral comparison between LNE and METAS to validate the calibration method described in the manuscript. Picarro G2103 was the analyser that METAS had available for this comparison exercise. Therefore, the analyser is not a reference analyser, but an analyser use to compare LNE's and META's standards. To clarify this, we have added to the second sentence on section 7: "In March 2020, METAS calibrated their PICARRO G2103 gas analyser […] to use it as a transfer standard at LNE".  We have also modified the second sentence of the second paragraph of section 9 as follows: "The bilateral comparison organized by LNE and METAS used a PICARRO G2103 gas analyzer as transfer standard based on a robust spectroscopic method and set up in a lot of AASQA monitoring stations. Both institutes generated reference gas mixtures dynamically by two different reference generators and used the above-mentioned PICARRO as comparator."

It will be more interesting to give an example by applying this dynamic reference gas in the real course of ammonia measurements (e.g., show the time series in the field measurements by overlapping the calibration process on the ambient ammonia concentrations, based on figure 6).

The objective of this study was to develop a new reference for $NH_3$ at low amount fraction level and to validate it before using the reference for the calibration of monitoring station analysers. This new procedure has not been implemented in the routine of the monitoring stations yet. Therefore, we cannot show an example of measurement time series in the field by overlapping the calibration process on the ambient ammonia.

Technical corrections:

Lines 26-29: Some words or logic here looks not scientific sound, e.g., "raised great interest", "requests shows the interest" and "to guarantee".

We have addressed the Referee's concerns by modifying the text as follows: "We have modified the draft as follows: "The results highlighted the good agreement between the $NH_3$ reference generators developed by the two institutes and allowed to validate both LNE's reference generator and calibration procedure. Since the end of 2020, LNE calibrated several $NH_3$ analyzers from the French air quality monitoring networks (AASQA) using the newly developed SI-traceable RGMs. The enhanced number of calibrations provided, may promote and increase the comparability, accuracy and traceability of the $NH_3$ measurements carried out on the French territory."

Line 45: Some recent nitrogen isotopic study also suggested nonagricultural emissions of ammonia in urban regions, e.g., https://pubs.acs.org/doi/10.1021/acs.est.1c05884. Such findings signify the requirements in observing ammonia to reduce air pollution.

We have added this information to the manuscript: "However, recent studies based on isotopic nitrogen suggest that the contribution of non-agricultural sources to $NH_3$ emissions are greater than the agricultural ones in certain urban regions (Feng et al., 2022; Gu et al., 2022)."

Feng, S., Xu, W., Cheng, M., Ma, Y., Wu, L., Kang, J., Wang, K., Tan, A., Collett, J.L., Fang, Y., Goulding, K., Liu, X., Zhang., F. Overlooked nonagricultural and wintertime agricultural $NH_3$ emissions in Quzhou county, North China Plain: Evidence from $^{15}$N-Stable Isotopes, Environ. Sci. Technol. Lett., 9, 127–133, https://doi.org/10.1021/acs.estlett.1c00935, 2022

Gu, M., Pan, Y., Walters, W.W, Sun, Q., Song, L., Wang, Y., Xue, Y., Fang, Y. Vehicular Emissions Enhanced Ammonia Concentrations in Winter Mornings: Insights from Diurnal Nitrogen Isotopic Signatures, Environ. Sci. Technol., 56, 3, 1578–1585, https://doi.org/10.1021/acs.est.1c05884, 2022.

Line 127: Be consistent in the text and change sulfur dioxide, nitrogen dioxide to $SO_2$, $NO_x$.

We have corrected the manuscript to be consistent in the text.

Lines 508-509: The last sentence in the conclusion is not scientific sound.

We have addressed the Referee's concerns by removing the sentence and modifying the text as follows: "The application of these findings, together with the use of the newly developed $NH_3$ SI-traceable RGMs by the French air quality monitoring stations within AASQA, may promote the accuracy, comparability and traceability of $NH_3$ measurements carried out in France".